# The Sequence Dependent Nanoscale Structure of CENP-A Nucleosomes

**DOI:** 10.3390/ijms231911385

**Published:** 2022-09-27

**Authors:** Tommy Stormberg, Yuri L. Lyubchenko

**Affiliations:** Department of Pharmaceutical Sciences, University of Nebraska Medical Center, Omaha, NE 68198, USA

**Keywords:** centromere chromatin, CENP-A nucleosomes, atomic force microscopy, nanoscale structure of nucleosomes, alpha satellite DNA

## Abstract

CENP-A is a histone variant found in high abundance at the centromere in humans. At the centromere, this histone variant replaces the histone H3 found throughout the bulk chromatin. Additionally, the centromere comprises tandem repeats of α-satellite DNA, which CENP-A nucleosomes assemble upon. However, the effect of the DNA sequence on the nucleosome assembly and centromere formation remains poorly understood. Here, we investigated the structure of nucleosomes assembled with the CENP-A variant using Atomic Force Microscopy. We assembled both CENP-A nucleosomes and H3 nucleosomes on a DNA substrate containing an α-satellite motif and characterized their positioning and wrapping efficiency. We also studied CENP-A nucleosomes on the 601-positioning motif and non-specific DNA to compare their relative positioning and stability. CENP-A nucleosomes assembled on α-satellite DNA did not show any positional preference along the substrate, which is similar to both H3 nucleosomes and CENP-A nucleosomes on non-specific DNA. The range of nucleosome wrapping efficiency was narrower on α-satellite DNA compared with non-specific DNA, suggesting a more stable complex. These findings indicate that DNA sequence and histone composition may be two of many factors required for accurate centromere assembly.

## 1. Introduction

The centromere is a specialized segment of chromosomes that aids in chromosomal segregation. There are several key features of the centromere that likely lead to unique structures distinct from those found in bulk chromatin that possess dynamic properties necessary for proper centromere function. The distinction between the centromere and bulk chromatin appears at the nucleosomal level; in most eukaryotes, including humans, this is the presence of histone H3 variant CENP-A [1,2]. CENP-A has been shown to be necessary for proper kinetochore association and chromosome segregation [3]. Kinetochores must be able to interact directly with the centromere in order to connect centromere chromatin to microtubules; this interaction is necessary for faithful chromosome segregation [4]. In fact, CENP-A deposition on non-centromere genomic sites can result in recruitment of all kinetochore components and formation of a neocentromere [5,6]. While CENP-A protein shares high sequence identity with histone H3 [7,8], they have distinct characteristics. CENP-A nucleosomes exhibit a reduced wrapping efficiency compared with their H3 counterparts, wrapping 13 bp less at the entry-exit sites of the nucleosome core [9]. At the same time, CENP-A nucleosomes demonstrate a similar, or even enhanced, core stability compared with H3 nucleosomes [10,11]. Moreover, unlike H3 nucleosomes, CENP-A nucleosomes are formed exclusively at the centromere [12,13].

Another key feature of centromeric chromatin in humans is the presence of tandem repeats of α-satellite DNA motifs. The α-satellite motifs share high sequence identity, and all motifs are A-T rich and 171 bp in length [14]. These motifs form higher order repeats spanning thousands of bases, which together comprise the centromere region spanning several megabases [15]. Recently, the human centromere was sequenced, highlighting the important role of α-satellite arrays in the interaction and almost exclusive localization of CENP-A [16]. An important characteristic of α-satellite DNA is that it contains a 17 bp motif called the CENP-B box. This motif specifically binds centromere protein B (CENP-B), another protein that is thought to play a significant role in the assembly of the centromere [17]. On the other hand, the CENP-B protein is considered non-essential to the survival of its host organisms due to the results of knockout mice studies [18]. This further complicates the understanding of the role of DNA sequence and protein interaction in the assembly of the centromere.

In a previous study, we identified dynamic translocation properties of CENP-A nucleosomes and proposed a model explaining the effect of nucleosomal DNA shortening on such nucleosomes [11,19,20]. However, the experiments were performed on the non-physiological 601 motif characterized by the high affinity for the assembly of H3 nucleosomes [21]. The goal of this study was to reveal nanoscale structural characteristics of nucleosomes assembled on the α-satellite sequence. We utilized Atomic Force Microscopy (AFM) with the capability to characterize nucleosomes on the single molecule level with nanometer resolution to address this issue [22,23,24,25,26]. We assembled CENP-A nucleosomes on three different substrates: a substrate containing an α-satellite motif, a substrate containing non-specific DNA and nucleosomes assembled on the Widom 601 positioning motif to determine whether CENP-A has distinguishing structural characteristics between α-satellite and other DNA sequences. H3 nucleosomes were also assembled on α-satellite DNA for comparison. We end-labeled DNA with streptavidin [27] to differentiate between DNA ends. We did not reveal highly specific assembly of CENP-A nucleosomes, as observed for the 601 motif, on the α-satellite DNA. A similar effect was found for H3 nucleosomes on the same substrate. Moreover, we identified similarities between CENP-A nucleosomes assembled on the α-satellite DNA and those assembled on non-specific DNA. The data indicate other factors, rather than simply CENP-A or DNA sequence, dictate the assembly of the centromere.

## 2. Results

### 2.1. AFM Analysis of Nucleosomes Assembled on α-Satellite DNA Motif

CENP-A and H3 nucleosomes were assembled on the α-satellite DNA substrate shown in Figure 1A. The substrate features a centrally positioned α-satellite sequence containing the CENP-B box flanked on both ends by non-specific DNA. DNA was terminated with biotin on the far flank allowing for distinction between DNA ends on assembled nucleosomes after streptavidin labeling. Incubation with streptavidin was performed immediately prior to the deposition of nucleosome samples on mica functionalized with APS for AFM imaging.

We performed AFM imaging of the nucleosomes. Representative images of the nucleosome samples are shown in Figure 2. A representative image of assembled CENP-A nucleosomes can be seen in Figure 2A, while a representative image of assembled H3 nucleosomes can be seen in Figure 2B. Nucleosome cores are identifiable as the bright globular features along the DNA substrate, while streptavidin is distinguishable as the smaller, less pronounced round features at the end of DNA substrates. From these images we measured the contour length of all nucleosome flanks and the nucleosome core position relative to the streptavidin-labeled DNA end. The data for flank lengths can be seen in Appendix A, where it is shown that the flank length distribution for both types of nucleosomes is quite similar. The nucleosome flank lengths were subtracted from the known DNA substrate length of 410 bp to determine the wrapping efficiency of each nucleosome. The results were plotted as histograms, shown in Figure 3. Fitting a Gaussian curve to each histogram produced an average wrapping efficiency of 125 ± 20 bp and 146 ± 22 bp for CENP-A and H3 nucleosomes, respectively. These values are consistent with the expected wrapping efficiency of fully assembled nucleosomes.

Mapping of nucleosome positioning was performed utilizing the end labeling of our DNA substrate, and the results are shown in Figure 4. The gray regions of the maps represent the free DNA flanks of the nucleosomes. The blue regions indicate the DNA occupied by the nucleosome. The maps are overlayed with a transparent green region, indicative of the α-satellite motif on the DNA substrate. The transparent gray bar is representative of the position of the CENP-B box within the α-satellite motif. The results for CENP-A and H3 mapping are strikingly similar. A decrease in end positioning is seen for CENP-A nucleosomes; H3 nucleosomes are observed to position at the end of the DNA substrate in 18% of cases (*n* = 196), while CENP-A nucleosomes are end-positioned in only 9% of cases (*n* = 194). However, no direct preference for the CENP-B box or the α-satellite motif is observed for either nucleosome type. These data suggest that the α-satellite motif does not inherently influence the positioning of CENP-A nucleosomes at the centromere.

### 2.2. AFM Analysis of CENP-A Nucleosomes on the Widom 601 Positioning Sequence

In order to assess how CENP-A nucleosome structure is affected by the α-satellite motif, we assembled nucleosomes on the substrate containing the well-defined Widom 601 nucleosome positioning motif. A representative image of the nucleosome assembly can be seen in Figure 5A. We performed the same analysis on these nucleosomes as we did with those assembled on the substrate containing the α-satellite motif, and the results are shown in Figure 5B,C. The histogram in Figure 5B describes the wrapping efficiency for nucleosomes assembled on the 601 motif. The wrapping efficiency of nucleosomes on the 601 motif display an asymmetric distribution, with a major peak at 121 ± 18 bp and a shoulder corresponding to a minor peak at 154 ± 13 bp. A similar asymmetric distribution was observed in our previous publication for CENP-A nucleosomes, which was explained by the formation of DNA loops inside the nucleosomes. The free DNA flank measurements are shown in Appendix A. Nucleosomes assembled on the 601 motif vary more in the flank length of the streptavidin-labeled arm compared with the non-labeled arm, suggesting that the nucleosomes over wrap in an asymmetric manner on this substrate.

Mapping of the nucleosomes is shown in Figure 5C. Nucleosomes assembled on the 601 substrate display consistent positioning along the central region of the DNA substrate; this is consistent with the position of the 601 motif, indicating that CENP-A nucleosomes bind tightly to this region. No such positioning effect was observed on the α-satellite substrate, highlighting the strong preferential binding of the 601 sequence and suggesting that CENP-A assembly along the α-satellite substrate does not behave as a sequence specific motif for the nucleosome assembly.

### 2.3. AFM Analysis of CENP-A Nucleosome on Non-Specific DNA

To assess whether any sequence-dependent wrapping or positioning effect is present in the α-satellite substrate, we assembled CENP-A nucleosomes on the DNA substrate containing no positioning sequence or centromeric sequence seen in Figure 1C. This substrate is termed the non-specific substrate. A representative image of nucleosomes on this substrate is shown in Figure 6A. The wrapping efficiency of nucleosomes assembled on non-specific DNA was determined and plotted as a histogram, as done before. The histogram, shown in Figure 6B, displays a broad Gaussian fitting with a peak centered around 128 ± 40 bp. This value is both larger and broader than that seen on other substrates. Mapping of CENP-A nucleosomes assembled on the non-specific DNA, shown in Figure 6C, revealed that the nucleosomes do not display preferential positioning. This trend is similar to that seen for nucleosomes assembled on the α-satellite motif, indicating that the α-satellite motif acts as non-specific DNA with regard to the positioning of nucleosomes at the centromere.

## 3. Discussion

In this study we explored the effect of DNA sequence on the assembly of CENP-A nucleosomes. CENP-A and H3 nucleosomes are both present at the centromere, where they wrap tandem repeats of α-satellite DNA. However, how the presence of both nucleosomes at the centromere and how their interaction with α-satellite DNA affects the structure and function of the centromere is still unknown. To our knowledge, this is the first direct comparison between CENP-A and H3 nucleosomes assembled on α-satellite DNA at the single-molecule level. We discovered that both nucleosomes assemble rather similarly along α-satellite DNA. Moreover, our characterization of CENP-A nucleosomes on different DNA substrates revealed that α-satellite DNA acts more as non-specific DNA with regard to nucleosome assembly than as a positioning motif. Below we discuss these findings and their implications in more detail.

The centromere is defined by the presence of α-satellite repeats and CENP-A nucleosomes. Yet, our results indicate that CENP-A nucleosomes treat α-satellite DNA similar to non-specific DNA. One important feature found in nearly all variations of α-satellite DNA in humans is the presence of the CENP-B box, the 17 bp motif that binds CENP-B protein [28]. It is important to consider that this motif could play a large role in the centromere structure as well. Our studies showed no specific preference for the α-satellite region for neither CENP-A nor H3 nucleosomes. However, when breaking down nucleosomes into sub-populations based on wrapping efficiency, we noted an interesting effect. We separated both CENP-A and H3 nucleosomes assembled on the α-satellite substrate into sub-populations based on how many turns the DNA wraps around the nucleosome core, and these can be seen in Figure 7 and Figure 8. In both figures, nucleosomes were separated into four groups- those that wrap less than 1.25 turns of DNA (Figure 7A and Figure 8A), those that wrap 1.25–1.5 turns (Figure 7B and Figure 8B), those that wrap 1.5–1.75 turns (Figure 7C and Figure 8C), and those that wrap more than 1.75 turns (Figure 7D and Figure 8D). Next to each map is a representative zoomed in snapshot of a nucleosome adopting that conformation. The blue regions in the maps indicate wrapped DNA. The green regions indicate the position of the α-satellite sequence, and the light gray line indicates the CENP-B box. We found that, as nucleosomes take on a more wrapped conformation, they tended to over wrap in the direction of the CENP-B box. If nucleosomes indeed over wrap toward the CENP-B box, it implicates the motif in both the positioning and stability of nucleosomes formed at the centromere. Future studies involving the CENP-B box and CENP-B protein are warranted to evaluate their effect on centromere structure and function.

We know that CENP-A is required for the faithful assembly of the kinetochore [3], and that CENP-A nucleosomes are found in high abundance at the centromere, so there must be important characteristics of the nucleosome structure on α-satellite that facilitate this process. It should be noted, however, that other factors can be responsible for these unique properties of the centromere, and these factors can serve as intermediaries between the DNA sequence and the CENP-A core. For example, recent studies have tested the importance of other centromere proteins, namely CENP-B and CENP-C, in a complex feedback loop regulating the faithful deposition and assembly of CENP-A at the centromere [29]. It may be that these proteins are critical in the positioning of CENP-A nucleosomes along the DNA substrate and the subsequent kinetochore assembly. These factors could explain the lack of preferential positioning observed in our system utilizing CENP-A nucleosomes and α-satellite DNA.

Another important feature to consider is that nucleosomes assemble in arrays along the centromere. While positional preference was not found in this study, CENP-A nucleosomes assembled on α-satellite DNA were identified as having the most stable wrapping efficiency, wrapping 125 ± 20 bp of DNA, shown in Figure 3. The commonly accepted value for a fully wrapped CENP-A nucleosome is 121 bp due to the 13 bp terminal DNA segments detached from the histone core [20]. CENP-A nucleosomes on other substrates were found more commonly in varied and over wrapped states, as seen in Figure 6. Previous studies have indicated the importance of internucleosomal interactions in canonical nucleosomes to their positioning and stability [24,25], and this effect may be even more critical at the centromere. The improved stability of CENP-A nucleosomes on α-satellite DNA could provide a synergistic effect with internucleosomal interactions, allowing this region to withstand the forces the centromere is subjected to during cell division [30]. Future studies are warranted to explore this property of CENP-A on α-satellite. Taken together, the results reported here suggest that assembly of nucleosomes at the centromere is dependent on a complex system of interactions, including interactions between CENP-A, α-satellite DNA, and other centromere proteins.

## 4. Materials and Methods

### 4.1. DNA Substrates

The DNA substrates used in nucleosome assembly were generated using PCR with a pUC57 plasmid vector from BioBasic (Markham, ON, Canada). A biotinylated reverse primer (IDT, Coralville, IA, USA) was used on all substrates for streptavidin labeling. Substrates containing the Widom 601 motif and the non-specific DNA totaled 377 bp in length, while the substrate containing the α-satellite motif totaled 410 bp in length. Schematics of the substrate designs can be seen in Figure 1. DNA sequences are listed as Appendix A. DNA substrates were separated by gel electrophoresis using 1% SeaKem LE Agarose gel (Lonza Group AG, Basel, Switzerland). The bands were excised and purified using QIAquick Gel Extraction Kit (Qiagen, Hilden, Germany). DNA concentration was then determined using NanoDrop Spectrophotometer ND-1000 (Thermo Fischer, Waltham, MA, USA).

### 4.2. Nucleosome Reconstitution and Streptavidin Labeling

For CENP-A nucleosome studies, recombinant human CENP-A/H4 tetramers and H2A/H2B dimers were purchased from EpiCypher (Research Triangle Park, NC, USA), while for H3 nucleosomes, recombinant human octamers containing H3 were purchased from EpiCypher. Nucleosome assembly was achieved using the continuous dilution method, as described in [31]. Briefly, purified DNA was mixed with histones at a 1:2:1 ratio of DNA:dimer:tetramer for CENP-A and a 1:1 ratio of DNA:octamer for H3 nucleosomes in a buffer containing 2 M NaCl and 10 mM Tris pH 7.5. A syringe pump was used to reduce the concentration to 200 mM NaCl over the course of 120 min. Nucleosomes were then stored at 4 °C.

Before deposition for imaging, nucleosomes were labeled at a terminal biotin using streptavidin, which binds specifically to the biotinylated DNA terminus [32]. Assembled nucleosomes were incubated with streptavidin for 10 min at a molar ratio of 2:1 streptavidin:nucleosome in incubation buffer (10 mM Tris pH 7.5, 125 mM NaCl, 5 mM MgCl_2_). After incubation, samples were immediately prepared for imaging as described below.

### 4.3. Atomic Force Microscopy Imaging

Sample preparation for AFM imaging was performed as previously described [24]. Freshly cleaved mica was functionalized with a solution of 1-(3-aminopropyl)- silatrane (APS). The nucleosome stock solution was diluted from 300 nM to 2 nM in imaging buffer (10 mM HEPES pH 7.5, 4 mM MgCl_2_) immediately before deposition on the functionalized mica. The sample was left to incubate for 2 min before being rinsed with water and dried with argon flow. Samples were stored in vacuum before being imaged on a Multimode AFM/Nanoscope IIId system using TESPA probes (Bruker Nano Inc., Camarillo, CA, USA). A typical image captured was 1 × 1 μm in size with 512 pixels/line.

### 4.4. Data Analysis

DNA contour length analysis was performed by measuring from the center of the protein label to the free end of DNA using Femtoscan software (Advanced Technologies Center, Moscow, Russia). Flank measurements for the nucleosomes were obtained by measuring from the center of the protein label to the center of the nucleosome for the labeled arm and from the free end of DNA to the center of the nucleosome for the unlabeled arm. 5 nm was subtracted from each measured flank length to account for the size contributed by the histone core [33,34]. Free DNA was measured on each image. The mean value of the free DNA was then divided by the known length of a given substrate. The resulting value was used as a conversion unit. All other measurements on each image were divided by the calculated conversion unit to convert measurements in nm to base pairs (bp).

Wrapping efficiency was calculated by subtracting the combined flank lengths from the known DNA lengths as done previously [24,25]. These methods were used to produce histograms and mapping of nucleosome position and wrapping efficiency. Subsets of these data were sorted by wrapping efficiency to compare nucleosome structure at various wrapping states. All graphs were created using Origin, Version 6.0 and Origin 2016 (OriginLab Corporation, Northampton, MA, USA). Gaussian fitting was performed automatically using Origin software’s “Fit Gaussian” analysis function on each histogram. The reported peak values correspond to the output of the “Fit Gaussian” analysis function.

## Figures and Tables

**Figure 1 ijms-23-11385-f001:**
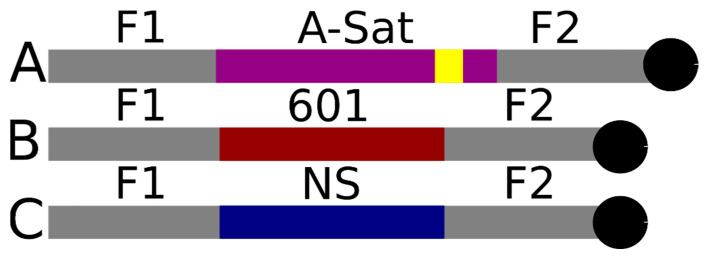
Schematic representation of the substrates used. Schematic (**A**) represents the α-satellite substrate. Schematic (**B**) represents the 601 substrate. Schematic (**C**) represents the non-specific substrate. F1 is 114 bp and F2 is 125 bp on substrate (**A**). F1 is 113 bp and F2 is 117 bp on substrates (**B**,**C**). A-sat represents the α-satellite sequence of 171 bp; the yellow region represents the 17 bp CENP-B box within the α-satellite sequence; 601 represents the Widom 601 sequence of 147 bp; NS represents a non-specific sequence of 147 bp; the black circles represent the biotin label at the end of each template.

**Figure 2 ijms-23-11385-f002:**
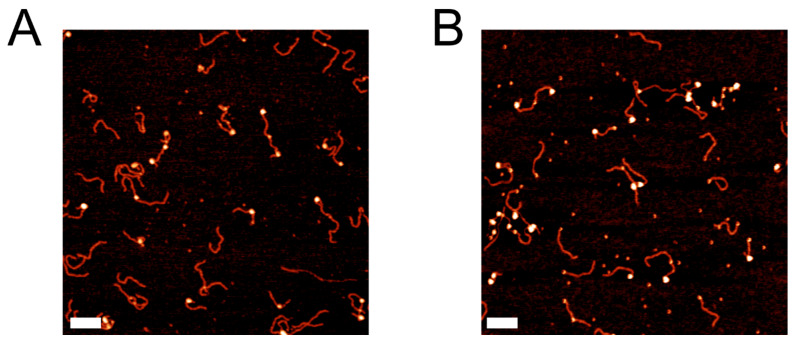
Representative images of CENP-A nucleosomes (**A**) and H3 nucleosomes (**B**) assembled on the α-satellite DNA substrate labeled with streptavidin. Scale bars indicate 100 nm.

**Figure 3 ijms-23-11385-f003:**
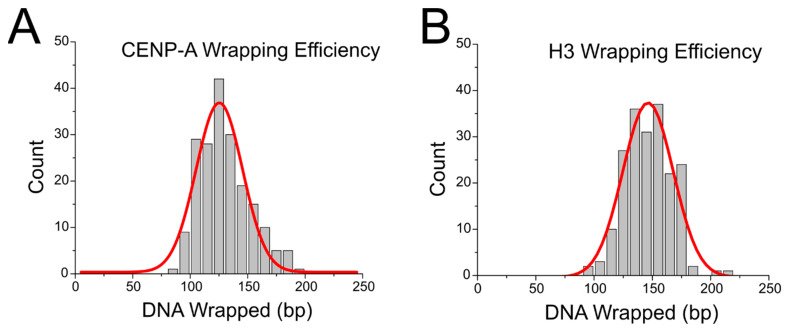
Histograms with Gaussian fitting of CENP-A nucleosome (**A**) and H3 nucleosome (**B**) wrapping efficiency on the α-satellite DNA substrate reveal a mean wrapping efficiency of 125 ± 20 bp and 146 ± 22 bp for CENP-A and H3 nucleosomes, respectively.

**Figure 4 ijms-23-11385-f004:**
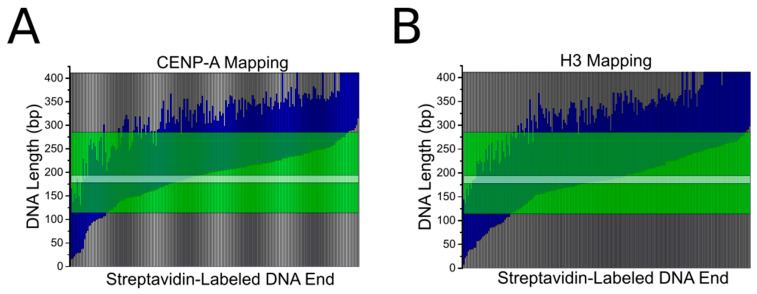
Mapping of CENP-A (**A**) and H3 (**B**) nucleosome position along the α-satellite DNA substrate. Gray bars represent free DNA flanks. Blue regions indicate DNA wrapped by the nucleosome. The transparent green region indicates the α-satellite motif. The transparent white bar indicates the position of the CENP-B box within the α-satellite motif. Nucleosome position is organized by smallest to greatest distance from the streptavidin-labeled DNA end.

**Figure 5 ijms-23-11385-f005:**
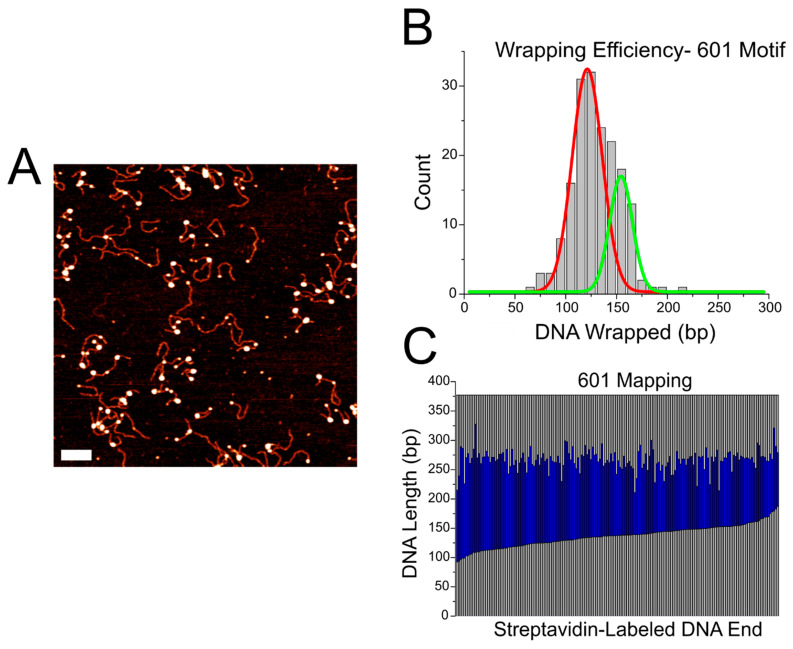
Analysis of CENP-A nucleosomes assembled on the DNA substrate containing the 601 motif. (**A**) Representative image of nucleosome samples. Scale bar indicates 100 nm. (**B**) Histogram of wrapping efficiency displays a bimodal distribution, with a fully wrapped major peak at 121 ± 18 bp, as indicated by the red line, and an over wrapped minor peak at 154 ± 13 bp, as indicated by the green line. (**C**) Mapping of nucleosomes assembled on the 601 motif display central positioning in line with the position of the motif.

**Figure 6 ijms-23-11385-f006:**
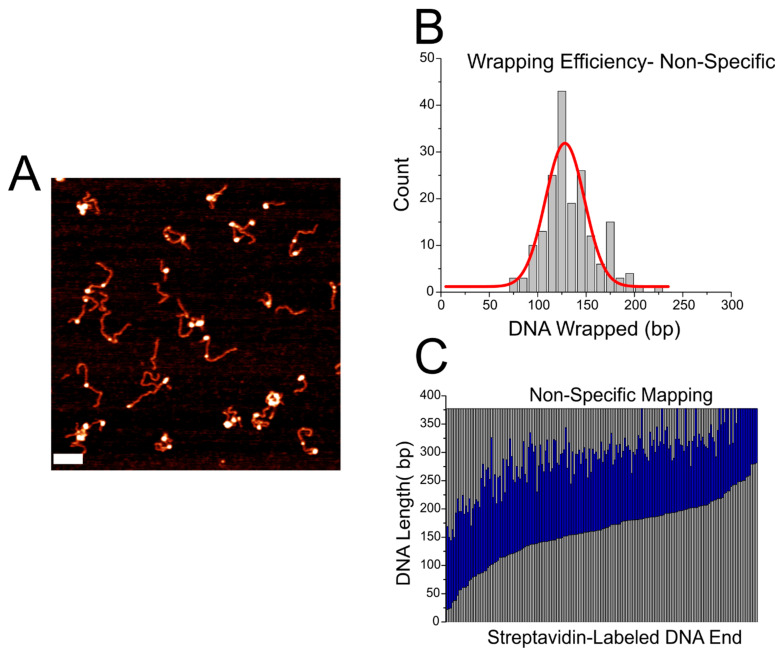
Analysis of CENP-A nucleosomes assembled on the non-specific DNA substrate. (**A**) Representative image of nucleosome samples. Scale bar indicates 100 nm. (**B**) Histogram of wrapping efficiency displays a broad Gaussian fitting with a peak centered around 128 ± 40 bp. (**C**) Mapping of nucleosomes assembled on non-specific DNA display no preferential positioning.

**Figure 7 ijms-23-11385-f007:**
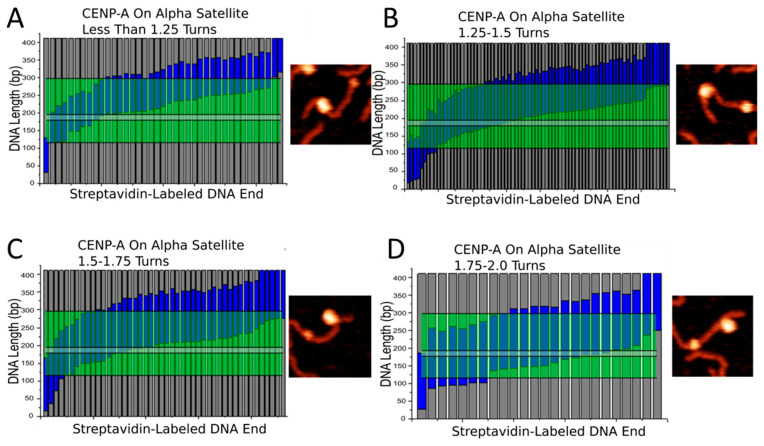
Mapping of CENP-A nucleosome position on the α-satellite substrate based on wrapping efficiency. Each bar denotes a single nucleosome. Dark gray regions indicate free DNA flanks. Blue regions indicate DNA wrapped around nucleosome core. Green regions indicate the position of the α-satellite motif. Thin gray line indicates position of the CENP-B box. Representative image of nucleosome with corresponding wrapping efficiency shown to the right of each map. (**A**) Under wrapped nucleosomes. (**B**) Fully wrapped nucleosomes. (**C**,**D**) Over wrapped nucleosomes.

**Figure 8 ijms-23-11385-f008:**
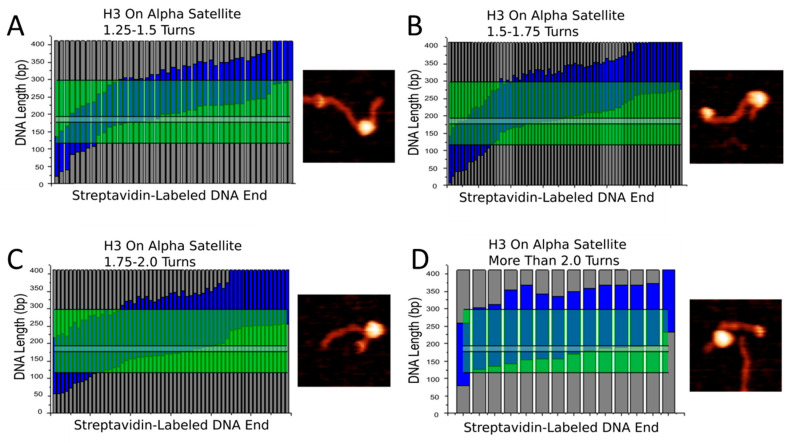
Mapping of H3 nucleosome position on the α-satellite substrate based on wrapping efficiency. Each bar denotes a single nucleosome. Dark gray regions indicate free DNA flanks. Blue regions indicate DNA wrapped around nucleosome core. Green regions indicate the position of the α-satellite motif. Thin gray line indicates position of the CENP-B box. Representative image of nucleosome with corresponding wrapping efficiency shown to the right of each map. (**A**) Under wrapped nucleosomes. (**B**) Fully wrapped nucleosomes. (**C**,**D**) Over wrapped nucleosomes.

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
