# Peer review of "The Sequence Dependent Nanoscale Structure of CENP-A Nucleosomes"

_ijms, 2022, doi:10.3390/ijms231911385_

Round 1
Reviewer 1 Report
In this paper, the authors studied nucleosomes assembled on ?-satellite sequence through Atomic Force Microscopy (AFM) in a nanoscaled manner. They prepared three different DNA substrates with CENP-A assembled nucleosomes: ?-satellite motif; nonspecific DNA motif and 601 positioning motifs with higher histone H3 affinity. They have end-labeled DNA with streptavidin to differentiate between DNA ends and observe the DNA length through AFM.
The technique is interesting for detecting nucleosome assembly in a nanoscale manner. In Figures 1-3, they have compared the mapping distance of CENP-A and H3 nucleosomes assembled on ?-satellite motif. Since the ?-satellite sequence containing the CENP-B box that is responsible for CENP-B binding, they also analyzed the CENP-B positioning on the assembled nucleosomes. However, with similar wrapping efficiency, they did not detect a preference for CENP-B or ?-satellite motif that is responsible for CENP-A nucleosomes' position at the centromere.
In Figure 4, they have tested CENP-A nucleosomes assembled on 601 positioning sequence. They detected asymmetric wrapping distribution of CENP-A nucleosomes, and the wrapping focused on the central region of nucleosomes. This shows CENP-A assembled along the a-satellite, but this assembly is not specifically targeting the ?-satellite motif.
In Figure 5, the authors analyzed CENP-A nucleosome assembly on non-specific DNA. The non-specific mapping shows a similar pattern to the ?-satellite motif with larger and broader wrapping distribution than other substrates. However, this result shows ?-satellite motif acts as non-specific DNA concerning the positioning of nucleosomes at the centromere.
Interestingly, they separated nucleosome assembly on ?-satellite motif into groups based on their wrapping turns. They found for those nucleosomes to take on more wrapped conformation, they tended to overwrap in the direction of the CENP-B box on the ?-satellite motif. With this evidence, they implied other factors can contribute to the CENP-A positioning at the centromere, and more studies can be done on CENP-B or CENP-C.
There are several typos in the article and missed figure labels in Figure7. The blue color label for the DNA occupied by nucleosomes can be more distinctive from the background color. The paper contains innovative ideas and established AFM as the technology to study nucleosome assembly in a nanoscale manner that can be expanded to other research, although it failed to prove the significance of the ?-satellite motif concerning the nucleosome positioning.
Author Response
There are several typos in the article and missed figure labels in Figure 7.
Response: We thank the reviewer for their comments. The article has been thoroughly proofread. Figure labels are present on Figure 7.
The blue color label for the DNA occupied by nucleosomes can be more distinctive from the background color.
Response: The background color has been brightened to make blue more distinctive in all mapping Figures 4-6 to make blue more distinctive and to better match Figures 7 and 8.
The paper contains innovative ideas and established AFM as the technology to study nucleosome assembly in a nanoscale manner that can be expanded to other research, although it failed to prove the significance of the ?-satellite motif concerning the nucleosome positioning.
Response: We thank the reviewer for their comments. We assert that the lack of positioning by the ?-satellite motif is an important finding itself, as it directs research to the other centromeric factors that can dictate the structure of the centromere.
Reviewer 2 Report
Scientists have been intensively studying the assembly of the nucleosome and DNA sequences that position nucleosomes for more than a quarter of a century. Of particular interest are nucleosomes assembled at the centromere, in which the canonical histone H3 is replaced by a specific centromeric one. In humans, this histone is called CENP-A. In this work, the authors used Atomic Force Microscopy to study in vitro the effect of DNA sequencing on the assembly and position of nucleosomes. The authors obtained new data and raised interesting questions for further research. Judging by previous publications, the authors have many years of research experience on this topic. The work is interesting, the experiments are logical and well-reasoned.
I have a few suggestions and minor clarifications.
I suggest moving Fig. S1A into the main text. Firstly, this allows the reader to immediately see the scheme of experiments, and secondly, the authors quite often refer to this figure in the text.
Since the paper is of interest to a wide range of readers, including those involved in the study of plant centromeres, to explain that CENP-A is a human centromeric histone,
Line 262-263 Authors contribution: replace Y.Y with Y.L
Author Response
I have a few suggestions and minor clarifications.
I suggest moving Fig. S1A into the main text. Firstly, this allows the reader to immediately see the scheme of experiments, and secondly, the authors quite often refer to this figure in the text.
Response: We thank the reviewer for their comments. Fig. S1 has been moved to the main text and is now Fig. 1. All other Figures have been adjusted accordingly.
Since the paper is of interest to a wide range of readers, including those involved in the study of plant centromeres, to explain that CENP-A is a human centromeric histone,
Response: We have clarified in the text that CENP-A is found in most eukaryotes, including humans, Line 9, Line 29, and Line 39.
Line 262-263 Authors contribution: replace Y.Y with Y.L
Response: This has been corrected.